# Towards an Optimal IPO Mechanism

**Fred E. Huibers**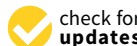

Faculty of Business and Economics, Amsterdam University of Applied Sciences,
1102 CV Amsterdam, The Netherlands; f.e.huibers@hva.nl

**Abstract:** Concerns about the negative consequences of the excessive underpricing of the current arrangement in the initial public offering (IPO) market for the provision of entrepreneurial finance—book building—have led to research into the viability of auctions for IPO pricing and allocation. IPO firms face a trade-off between the benefit of accurate and reliable IPO price discovery and the cost of underpricing. The main aim of this paper was to gain new scientific knowledge about this trade-off by measuring the impact of two key variables on this trade-off: capacity restraint and discount on the auction clearing price. Using controlled experiment methodology in multi-unit uniform price auctions we found that the most capacity-restricted auctions that also offer investors a discount are likely to produce the most accurate and reliable price discovery and consequently, the most predictable auction outcome. There are indications that a discount of 8% may suffice to incentivize investors to reliably contribute to price discovery. The resulting underpricing (and its variability) of these auctions is likely to be significantly lower than if book building would be used to price and allocate IPOs. Technological innovation in the IPO market through the application of recent advances in data science, experimental economics and artificial intelligence allows for the optimization of IPO mechanisms and crowdfunding platforms which in turn improves the access to equity required for entrepreneurial finance.

**Keywords:** entrepreneurial finance; IPO; auction; price discovery; underpricing

## 1. Introduction

The optimal mechanism for pricing and allocating initial public offerings (IPOs) has generated considerable debate among scholars and practitioners. The focus of the debate is on the role of intermediaries in the process of price discovery.

It is generally accepted that the accurate valuation of IPOs requires specialized knowledge and experience due to the riskiness of unseasoned equity issues of relatively young, small and innovative firms. Consequently, models explaining IPO (under)pricing assume that there is a considerable degree of information asymmetry among parties involved in the valuation and that lowering the information asymmetry improves the accuracy of IPO pricing (Ljungqvist 2007). The process of lowering the information asymmetry is commonly referred to as price discovery, the objective of which is to maximize the contribution of privately held information by the most informed investors[1] to IPO valuation in order to produce an outcome closest to intrinsic value[2] (Schnitzlein et al. 2019;

---

[1] This private information "on management quality, strategy, and the ability to outperform competitors" of an IPO firm is assumed to be held by institutional investors that "spend their careers weighing the conflicting claims and forecasts" (Sherman 2005).

[2] The intrinsic value (often also referred to as fair value) of a firm that goes public is eventually revealed in the secondary market as public information is incorporated in the stock price, but is not fully revealed in the primary market given the limited number of participants that set the price in that market.

Lowry et al. 2010; Sherman 2005; Wilhelm 2005; Biais and Faugeron-Crouzet 2002; Biais et al. 2002).
The method currently used to achieve this objective is controversial.

The internationally dominant method is book building. In book building, investment banks act as the agent of the IPO candidate—the principal. The banks select and invite a limited number of large institutional investors[3] to submit bids for IPO shares within a price range which the banks have set in advance. The criteria applied to select the investors that are allowed to bid are not published. Based on the solicited demand curve, the banks generally persuade IPO firms to set the uniform offer price significantly below the clearing price (i.e., the highest price at which all the shares on offer could be sold) and "leave money on the table" for the investors (Ljungqvist 2007). The banks do not publicly disclose the clearing price nor the discount applied to it. As the number of shares demanded above the offer price usually exceeds the supply, the banks allocate the shares by rationing. They typically make three categories with each a different—but within the category—allocation ratio. The allocation criteria are not published. After the shares have been allocated, public trading results in a first day closing price that—on average—is about 15–20% higher than the offer price[4] (Wilhelm 2005; Ljungqvist 2007). This price differential is commonly referred to as underpricing.

In book building, investment banks are granted discretionary power in the pricing and allocation of the shares by the principal, the IPO firm. The controversy surrounding book building is about the use of these powers: are they optimally used for price discovery in the interest of IPO firms or is that not the case due to a severe principal–agent problem? Are investment banks acting as neutral agents that limit underpricing to what is needed to reward informed investors as modelled by Benveniste and Spindt (1989) or is the investment ban the case due to a severe principal–agent problem? Are investment banks acting as neutral agents that are prone to excessively underprice IPOs for their own benefit? Ritter (2011) concludes from his extensive review of the IPO literature that a severe principal–agent problem exists which originates from the oligopolistic market structure of investment banking and that it results in excessive underpricing in equilibrium which in turn increases the cost of equity and lowers firm access to the IPO market[5].

The importance of economic growth, innovation and the job creation to increasing firm access to the IPO market is recognized by economic policy makers. The European IPO Task Force acknowledges that for firms "the failure of the IPO market to facilitate their access to capital markets hampers their growth and lowers potential employment" and that "benefits of well-functioning IPO markets accrue to the whole economy" (De Backer 2015). Similar views led to the introduction of the Jumpstart Our Business Startups Act (JOBS Act) to improve access to the US IPO market as a source of funding for innovative small businesses. However, the evidence on the impact of the JOBS Act on the functioning of the IPO market is mixed. Dambra et al. (2015) finds evidence that the JOBS Act has had a positive impact on the functioning of the IPO market while others do not (Chaplinsky et al. 2017). The mixed evidence suggests that there is room for improvement on a more fundamental level: changing the IPO process itself. This study contributes to the research dedicated to the optimization of the IPO mechanism in order to improve access to funding for young innovative companies that are important contributors to economic growth and employment.

---

[3]　Depending on the size of the IPO, only about 100–200 institutional investors are invited as they expect the investment banks to set up a physical meeting with the top management of the IPO firm. These road shows are limited to a maximum of ten business days, thereby constraining the number of participants in book building.

[4]　Ljungqvist (2007, page 378) notes: "Since the 1960s, this 'underpricing discount' has averaged around 19% in the United States, suggesting that firms leave considerable amounts of money on the table."

[5]　Ritter (2011) also presents evidence of the occurrence of what he refers to as CLAS controversies: "C is the payment of excessive commissions by investors as a way of currying favor for IPO allocations. L is laddering, the practice of allocating shares in return for promises of additional purchases once the stock starts trading. A is biased analyst recommendations, with underwriters competing for business from issuers by either implicitly or explicitly promising favorable coverage from their research analysts. S is spinning, the practice of allocating shares from other IPOs to the personal brokerage accounts of issuing firm executives in return for investment banking business from the executives' company."

Concerns about the negative consequences of the current arrangements in the IPO market for the provision of entrepreneurial finance have led to research into the viability of auctions as the market-driven alternative for price discovery that does not suffer from the principal–agent problem. Biais and Faugeron-Crouzet (2002), Sherman (2005), Jagannathan and Sherman (2005), and Jagannathan et al. (2015) showed that standard sealed bid multi-unit common value uniform price auctions (hereafter: standard auctions) are handicapped relative to book building in the process of price discovery and that this explains why book building is—to date—the IPO pricing and allocation method of choice despite the principal–agent problems that surround it.

Institutional (informed) investors that are invited to bid in book building know that there are only a limited number of peers in the price-discovery process and that they will be rewarded with a payoff consisting of an allocation of shares that are discounted by 15–20% on average relative to the market price. Conversely, in a standard auction, informed investors are considerably more uncertain about the number and behavior of competing bidders and the payoff that they can expect from participation (Jagannathan et al. 2015). As all investors are treated equally in the pricing and allocating of the shares, the impact of free riding behavior of less informed investors[6] on the expected payoff of informed investors is much higher in auctions than in book building, causing informed investors to not fully contribute to price discovery by shaving their bids and/or limiting their participation (Sherman 2005).

In order to maximize the informed investor contribution to price discovery, standard auctions need to be modified to manage investor access (Sherman 2005) and incentivize informed investors to contribute their private information (Biais and Faugeron-Crouzet 2002). Excluding retail investors (Jagannathan and Sherman 2005; Jagannathan et al. 2015) and imposing capacity constraints on informed investors (Schnitzlein and Shao 2013) increases the contribution to price discovery of informed investors by lowering the impact of free riding on the auction clearing price. However, Schnitzlein and Shao (2013) found that—even after maximizing the number of shares that informed investors may bid for—free riding remains prevalent with around half the bids submitted above the price signal of intrinsic value. They concluded that imposing capacity constraints—while contributing to price discovery—needs to be complemented with incentives for the informed investors to contribute their private information. Jagannathan et al. (2015) draw the same conclusion from a comparative review of international IPO practices. The need to incentivize informed investors to contribute their private information in auctions by offering them a discount to the auction clearing price is theoretically derived and empirically confirmed by Biais and Faugeron-Crouzet (2002).

Therefore, IPO firms face a trade-off. Maximizing the contribution of private information to price discovery in order to achieve a clearing price that is closest to the intrinsic value of the IPO shares (i.e., maximizing the valuation accuracy that is instrumental in the generation of reliable and realistic proceeds[7]) comes at the cost of setting the offer price at a discount to the clearing price and imposing capacity constraints on investors that lower IPO firm proceeds in the form of underpricing. The optimal IPO mechanism generates the optimal results of this trade-off (Shim et al. 2019; Schnitzlein and Shao 2013; Sherman 2005; Derrien and Womack 2003; Biais and Faugeron-Crouzet 2002; Biais et al. 2002).

The main aim of this paper was to gain new scientific knowledge about this trade-off by using a controlled experiment to measure the impact of two key variables on this trade-off: capacity restraint and discount on the auction clearing price.

**Hypothesis 1.** *We hypothesized that discounts significantly improved IPO valuation accuracy and reliability but at the expense of more underpricing.*

---

[6] Free riders put in unrealistically high bids in order to increase the probability of an allocation on the assumption that these bids will not inflate the offer price far above the price that will come about when IPO shares subsequently start trading on the stock market.

[7] This outcome is often referred to in the literature as efficient pricing (e.g., Schnitzlein and Shao 2013; Biais and Faugeron-Crouzet 2002) where the efficiency of price discovery is measured by both the deviation of the auction clearing price from the intrinsic value (i.e., valuation accuracy) and the standard deviation of this measure (i.e., reliability).

Starting from the standard auction, we first measured the effect of incentivizing informed investors with discounts on valuation accuracy, reliability and underpricing.

**Hypothesis 2.** *Second, we hypothesized that imposing capacity constraints significantly improve IPO valuation accuracy and reliability but at the expense of more underpricing.*

Therefore, we also measured the effect of imposing capacity constraints on valuation accuracy, reliability and underpricing.

**Hypothesis 3.** *Third, we measured the combined effect of offering a discount and imposing capacity constraints to test if there was a combined effect of the two key variables on valuation accuracy, reliability and underpricing.*

We find that informed investors need to be incentivized with a discount on the auction clearing price in order to participate. If they are not offered the discount, their expected payoff is negative. The accuracy and reliability of price discovery significantly improves when investors are incentivized but at the cost of significantly more underpricing. Imposing capacity constraints has a similar trade-off effect on auction outcomes. The combination of imposing capacity constraints and offering discounts in auctions has a positive effect on the valuation accuracy and reliability but at the cost of additional underpricing. This study thus makes an important and novel contribution to the academic literature about optimal IPO mechanisms by measuring the impact of both key variables on IPO valuation accuracy, reliability and underpricing by using a controlled experiment.

## 2. Experimental Design, Subjects and Procedures and Data

### 2.1. Experimental Design

The experiment consisted of nine treatments that were repeated 10 times resulting in 90 auctions. In each auction, 30 shares were sold to nine bidders, four of them large and five small. Before each auction began, the subjects were randomly assigned to being large or small and notified of their role. Each subject was required to submit two bids in each auction each consisting of the number of shares and a unique bid price that was an integer value from the interval [6,34].

The bidders competed in a common value auction where the intrinsic value of the shares V was an integer value that was randomly generated from a uniform distribution [12, 28] at the start of each auction but was not revealed to the subjects. Both large and small bidders were assumed to be informed institutional investors. Large investors were assumed to be better informed than small investors about V and were given a price signal[8] that was an integer value that was randomly generated from a uniform distribution [V − 3, V + 3] while the small investors were given a price signal that was randomly generated from a uniform distribution [V − 5, V + 5]. We employed the conventional pro rata on the margin rule, in which demand above the auction clearing price was fully served and only bids at the auction clearing price were rationed (Kremer and Nyborg 2004).

We design nine treatments along two dimensions: the discount on the auction clearing price offered to bidders and the capacity constraint imposed on bidders. As both the discount and the capacity constraint took on three discrete levels, this resulted in nine treatments. Table 1 provides an overview of the auction design features of the nine treatments.

---

[8] We assume that institutional investors are endowed with private information pertaining to IPO valuation (Ritter 2011; Rock 1986) and therefore—different from Schnitzlein et al. (2019)—we did not offer investors the choice to purchase the price signal to improve their valuation accuracy nor the choice of non-participation in the auctions. We note that Schnitzlein et al. (2019) found that there is no significant difference in both the rate of information purchase nor of participation across treatments. Both rates are close to 100%.

**Table 1.** Nine Treatments categorized on the discount and capacity constraint.

| | Capacity Constraint | | |
|---|---|---|---|
| **Discount** | **Low** | **Mid** | **High** |
| low | 1 | 2 | 3 |
| mid | 4 | 5 | 6 |
| high | 7 | 8 | 9 |

We set the low, mid and high discount at 0%, 8%, and 12% of the auction clearing price, respectively. The low, mid and high capacity constraint for the large investors was set at 18, 15, and 12 shares, respectively. Finally, we set the low, mid and high capacity constraint for small investors at 4, 3, and 2 shares, respectively. Each of the nine treatments had a unique combination of features. The auction used in treatment 5, for example, offered a discount of 8% on the auction clearing price and restricted large investors to bid for a maximum of 15 shares and restricted small investors to bid for a maximum of three shares.

*2.2. Subjects and Procedures*

On 8 October 2019, third- and fourth-year bachelor students following the FinTech course at the Amsterdam University of Applied Sciences were given a lecture on the use of book building and auctions for IPOs. On 11 October 2019, these students were invited by email to participate in an experiment in which they were to bid for IPO shares in an electronic auction. Nine students registered for the experiment that was scheduled to take place on 11 December 2019. None of the subjects had ever participated in an electronic auction of shares before.

At the beginning of the experiment, the subjects were given written instructions explaining how the unknown underlying intrinsic value V was generated, how V related to the price signal that the small and large investors received, how the auction clearing price was determined, and how the discount was applied to the auction clearing price and how to submit bids. They received the specific instruction that they had to submit two different bids in terms of price and quantity per auction and that the total quantity of the two bids could not exceed the maximum set for the auction. They were explained how to maximize their investment return consisting of the number of shares that they successfully bid for times the difference between the underlying V and the discounted auction clearing price. As the intrinsic value V is assumed to be equal to the closing price of the first trading day of the IPO, a positive (negative) outcome of the V minus the discounted clearing price was a profit (loss).

In addition to reading the instructions and offering subjects the opportunity to ask questions, the experimenter explained the structure of the Excel file used to perform the auctions. The subjects were explained where on their computer screen to submit their two bids electronically and where they could find the maximum number of shares they were allowed to bid for, the discount, and the price signal. The subjects were informed that they were to bid in 90 consecutive auctions starting from ten iterations of treatment 1, ten iterations of treatment 2, and repeating these steps until finishing with the tenth iteration of auction 9. After the experimenter verified that all the subjects understood the procedure and that they were not allowed to communicate amongst each other during the bidding, each subject was assigned to a different computer terminal in a room at the university. The bidding process was completed in approximately three hours.

*2.3. Data*

Each subject provided two bids in each of the 90 auctions resulting in 180 bids per subject. Therefore, the nine subjects provided 1620 bids. For each auction, the following variables were calculated: auction clearing price, pricing error, allocations to large and small investors, payoff, free riding frequency and underpricing. To illustrate the calculation of the variables we used the illustrative input of one of the

ten auctions of treatment 5 in which the discount was set at 8%, the maximum number of shares that large (small) investors could bid for was 15 (3) and the intrinsic value was set at 26 (Table 2).

**Table 2.** Illustrative subject auction input.

| Investor | Identity | Price | Shares | Cumulative |
|----------|----------|-------|--------|------------|
| 1 | large | 29 | 11 | 11 |
| 2 | small | 28 | 2 | 13 |
| 1 | large | 27 | 4 | 17 |
| 2 | small | 26 | 1 | 18 |
| 3 | large | 25 | 11 | 29 |
| 4 | small | 24 | 2 | 31 |
| 3 | large | 23 | 4 | 35 |
| 4 | small | 22 | 1 | 36 |
| 5 | large | 21 | 10 | 46 |
| 6 | small | 20 | 2 | 48 |
| 5 | large | 19 | 5 | 53 |
| 6 | small | 18 | 1 | 54 |
| 7 | large | 17 | 9 | 63 |
| 8 | small | 16 | 1 | 64 |
| 7 | large | 15 | 6 | 70 |
| 8 | small | 14 | 2 | 72 |
| 9 | small | 13 | 1 | 73 |
| 9 | small | 12 | 2 | 75 |

The input shown in Table 2 is fictional and serves to illustrate the calculation of the variables. First, the bids were ranked from highest to lowest bid price (column "price"). Then, the cumulative number of shares of the bids was calculated (column "cumulative"). The auction clearing price was the price at which the cumulative volume equals 30, the total number of shares available. In this case the auction clearing price equaled 24 (italic number in column "price").

The pricing error (error) equaled the absolute difference between the auction clearing price of 24 and the intrinsic value of 26. In this case the pricing error equaled 2. Large investor 1 received an allocation of 15 shares (11 + 4), small investor 2 received an allocation of three shares (2 + 1), large investor 3 received an allocation of 11 shares, and small investor 4 receives an allocation of one share (half of what investor 4 bid for; all column "shares"). Therefore, the allocation to large investors equaled 26 (15 + 11) shares and the allocation to small investors equaled four (3 + 1) shares.

Payoff was calculated as follows:

$$payoff = allocation \times (intrinsic\ value - (auction\ clearing\ price - discount)) \tag{1}$$

In this example, the difference between the intrinsic value (26) and the discounted auction clearing price (0.92 × 24) was 3.92[9]. The payoff for the large (small) investors equaled 101.92 (15.68) since 26 and four shares which were allocated to them respectively and the profit per share equaled 3.92.

The free riding frequency (freq) was calculated as follows:

$$freq = number\ of\ shares\ with\ a\ bid\ price\ above\ signal/number\ of\ shares\ bid\ for \tag{2}$$

---

[9]    If the intrinsic value was 21, the investors would have suffered a loss of 1.08 per share.

Assuming that the signal price was equal to the intrinsic value of 26, in this example the total number of shares bid for equaled 75 (sum of all bids in column "shares") and the number of shares with a bid price above 26 equaled 17 (11 + 2 + 4 in the column "shares"). Therefore, the frequency of free riding in this auction equaled 22.667% (17/75).

Underpricing (UP) was calculated as follows:

$$\text{underpricing} = (\text{intrinsic value} - (\text{auction clearing price} - \text{discount}))/$$
$$(\text{auction clearing price} - \text{discount}) \tag{3}$$

In this example, underpricing equaled 17.754% ((26 − (0.92 × 24)/(0.92 × 24)).

Table 3 reports the descriptive statistics of the 90 auctions that the subjects completed in the experiment.

**Table 3.** Descriptive statistics.

| Variable | N | Mean | Std. Deviation | Minimum | Maximum |
|---|---|---|---|---|---|
| UP | 90 | 3.605% | 11.309% | −33.333% | 41.333% |
| payoff | 90 | 19.160 | 64.074 | −240.000 | 150.000 |
| freq | 90 | 52.758% | 11.888% | 22.222% | 78.000% |
| error | 90 | 1.444 | 1.415 | 0.000 | 8.000 |
| allocat_large | 90 | 22.056 | 3.406 | 10.000 | 29.000 |
| allocat_small | 90 | 7.944 | 3.406 | 1.000 | 20.000 |

## 3. Results

### 3.1. The Effect of Offering Discounts on Auction Outcomes

Table 4 shows the average underpricing, payoff, frequency of free riding, pricing error and allocations to large and small investors of auctions where investors were not offered a discount (the 30 auctions of treatments 1, 2 and 3) on the market clearing price of the auction compared to that of the auctions where investors were offered a discount (all other 60 auctions).

**Table 4.** Outcomes of auctions with or without a discount.

| Discount_Dummy * | | N | Mean | Std. Deviation | Std. Error Mean |
|---|---|---|---|---|---|
| UP | 0 | 30 | −3.340% | 10.357% | 1.891% |
| | 1 | 60 | 7.078% | 10.169% | 1.313% |
| payoff | 0 | 30 | −23.000 | 67.627 | 12.347 |
| | 1 | 60 | 40.239 | 50.886 | 6.569 |
| freq | 0 | 30 | 55.681% | 12.563% | 2.294% |
| | 1 | 60 | 51.296% | 11.361% | 1.467% |
| error | 0 | 30 | 1.633 | 1.712 | 0.313 |
| | 1 | 60 | 1.350 | 1.246 | 0.161 |
| allocat_large | 0 | 30 | 20.433 | 4.006 | 0.731 |
| | 1 | 60 | 22.867 | 2.758 | 0.356 |
| allocat_small | 0 | 30 | 9.567 | 4.006 | 0.731 |
| | 1 | 60 | 7.133 | 2.758 | 0.356 |

* Discount_dummy = 0 (1) for auctions without (with) a discount.

Offering an average discount of 10% resulted in higher underpricing (7.078% compared to −3.340%), a higher payoff (40.239 compared to −23.000), a lower frequency of free riding (51.296% compared to 55.681%) and a less pricing error (1.350 compared to 1.633).

The *t*-tests for the equality of the means reported in Table 5 showed that all the differentials were significant with the exception of the difference in the pricing error.

**Table 5.** Levene's test for the equality of variances and the *t*-test for the equality of the means of the outcomes of the auction with and without a discount.

| Variable | | Levene's Test Equality of Variances | | | *t*-Test Equality of Means | | | |
|---|---|---|---|---|---|---|---|---|
| | Assumption | F | Sig. | t | df | Sig. 2 Tail | Mean Diff. |
| UP | Equal variances assumed | 0.493 | 0.484 | −4.554 | 88 | 0.000 | −10.418% |
| | Equal variances not assumed | | | −4.526 | 57 | 0.000 | −10.418% |
| payoff | Equal variances assumed | 3.053 | 0.084 | −4.966 | 88 | 0.000 | −63.239 |
| | Equal variances not assumed | | | −4.522 | 46 | 0.000 | −63.239 |
| freq | Equal variances assumed | 1.277 | 0.262 | 1.666 | 88 | 0.099 | 4.385% |
| | Equal variances not assumed | | | 1.611 | 53 | 0.113 | 4.385% |
| error | Equal variances assumed | 1.711 | 0.194 | 0.894 | 88 | 0.374 | 0.283 |
| | Equal variances not assumed | | | 0.806 | 45 | 0.424 | 0.283 |
| allocat_large | Equal variances assumed | 3.117 | 0.081 | −3.376 | 88 | 0.001 | −2.433 |
| | Equal variances not assumed | | | −2.991 | 43 | 0.005 | −2.433 |
| allocat_small | Equal variances assumed | 3.117 | 0.081 | 3.376 | 88 | 0.001 | 2.433 |
| | Equal variances not assumed | | | 2.991 | 43 | 0.005 | 2.433 |

The results presented in Table 4 confirm that informed investors need to be incentivized to reliably contribute their private information to price discovery by offering them a discount. If the discount is not offered, the average payoff from participating in the auctions is negative, while it is positive in auctions that offer investors a discount on the market clearing price. In addition, the standard deviation of the payoff is significantly higher in auctions without a discount, making the reward significantly less predictable for investors (Tables 4 and 5).

The lower pricing error and free riding incidence in auctions with a discount reported in Table 4 are indications of the more reliable contribution to the price discovery of investors. As expected, the higher contribution to price discovery in the auctions that offer investors a discount is driven by the more informed, large investors. The average allocation (i.e., winning bids) to large investors in the auctions that offer a discount is significantly higher: 22.867 versus 20.433 shares out of the 30 shares that were allocated. In the discounted auctions, the winning bids of larger and more informed investors contribute 76.233% (22.867/30) to price discovery. This is only 68.110% in the auctions where no discount is offered. Moreover, the standard deviation of the allocations was significantly lower in the discounted auctions indicating a higher reliability of the price discovery process in discounted auctions (Table 5).

The larger role in the price discovery of better informed investors resulted in a lower pricing error and a lower frequency of free riding. This makes the resulting price discovery in auctions that offer a discount more predictable. However, IPO firms face a trade-off between underpricing and the reliability of IPO price discovery. Then, the impact of different levels of discount on auction outcomes is measured.

Table 6 shows the outcomes of auctions in which investors were offered a discount of 8% (the 30 auctions of treatments 4, 5 and 6) compared to the auction outcomes where investors were offered a discount of 12% (the 30 auctions of treatments 7, 8, and 9).

**Table 6.** Outcomes of auctions in which the investors are offered a discount of 8% or 12%.

| Midlow_Dummy * | | N | Mean | Std. Deviation | Std. Error Mean |
|---|---|---|---|---|---|
| UP | 1 | 30 | 6.439% | 12.567% | 2.294% |
| | 2 | 30 | 7.716% | 7.185% | 1.312% |
| payoff | 1 | 30 | 31.839 | 58.439 | 10.669 |
| | 2 | 30 | 48.640 | 41.315 | 7.543 |
| freq | 1 | 30 | 47.778% | 11.543% | 2.108% |
| | 2 | 30 | 54.815% | 10.185% | 1.859% |
| error | 1 | 30 | 1.47 | 1.383 | 0.252 |
| | 2 | 30 | 1.23 | 1.104 | 0.202 |
| allocat_large | 1 | 30 | 23.03 | 2.593 | 0.473 |
| | 2 | 30 | 22.70 | 2.950 | 0.539 |
| allocat_small | 1 | 30 | 6.97 | 2.593 | 0.473 |
| | 2 | 30 | 7.30 | 2.950 | 0.539 |

* Midlow_dummy = 1 (2) for auctions offering 8% (12%) discount.

The results presented in Table 7 indicate that there are no significant differences in the outcomes of auctions in which a discount of 8% was offered and the outcomes of auctions with a discount of 12%. The exception was the significantly lower frequency of free riding in auctions in which a discount of 8% was offered. These results implied that the lower level discount suffices to incentivize informed investors to contribute their private information reliably to price discovery.

**Table 7.** Levene's test for the equality of variances and the *t*-test for the equality of the means of the outcomes of the auction with a discount of 8% and 12%.

| | Levene's Test Equality of Variances | | | *t*-Test Equality of Means | | | |
|---|---|---|---|---|---|---|---|
| Variable | Assumption | F | Sig. | t | df | Sig. 2 Tail | Mean Diff. |
| UP | Equal variances assumed | 3.734 | 0.058 | −0.483 | 58 | 0.631 | −1.277% |
| | Equal variances not assumed | | | −0.483 | 46 | 0.631 | −1.277% |
| payoff | Equal variances assumed | 2.185 | 0.145 | −1.286 | 58 | 0.204 | −16.801 |
| | Equal variances not assumed | | | −1.286 | 52 | 0.204 | −16.801 |
| freq | Equal variances assumed | 0.052 | 0.820 | −2.504 | 58 | 0.015 | −7.037% |
| | Equal variances not assumed | | | −2.504 | 57 | 0.015 | −7.037% |
| error | Equal variances assumed | 2.165 | 0.147 | 0.722 | 58 | 0.473 | 0.233 |
| | Equal variances not assumed | | | 0.722 | 55 | 0.473 | 0.233 |
| allocat_large | Equal variances assumed | 1.198 | 0.278 | 0.465 | 58 | 0.644 | 0.333 |
| | Equal variances not assumed | | | 0.465 | 57 | 0.644 | 0.333 |
| allocat_small | Equal variances assumed | 1.198 | 0.278 | −0.465 | 58 | 0.644 | −0.333 |
| | Equal variances not assumed | | | −0.465 | 57 | 0.644 | −0.333 |

### 3.2. The Effect of Imposing Capacity Constraints on Auction Outcomes

Table 8 shows the average underpricing, payoff, frequency of free riding, pricing error allocations for the large and small investors of auctions in which the highest capacity constraint was imposed upon investors (the 30 auctions of treatment 3, 6 and 9) and of the other auctions that were less restrictive. Imposing the highest capacity constraint results in higher underpricing (10.245% compared to 0.285%), higher payoff (53.639 compared to 1.920), a lower frequency of free riding (49.444% compared to

54.415%), less pricing error (1.000 compared to 1.667) and a higher share of allocations to large investors (23.200 versus 21.483) compared to the other auctions.

**Table 8.** Outcomes of the auctions with the highest capacity constraint and other auctions.

| Constraint_Dummy * | | N | Mean | Std. Deviation | Std. Error Mean |
|---|---|---|---|---|---|
| UP | 0 | 60 | 0.285% | 11.085% | 1.431% |
| | 1 | 30 | 10.245% | 8.607% | 1.571% |
| payoff | 0 | 60 | 1.920 | 68.969 | 8.904 |
| | 1 | 30 | 53.639 | 32.825 | 5.993 |
| freq | 0 | 60 | 54.415% | 12.163% | 1.570% |
| | 1 | 30 | 49.444% | 10.756% | 1.964% |
| error | 0 | 60 | 1.667 | 1.580 | 0.204 |
| | 1 | 30 | 1.000 | 0.871 | 0.159 |
| allocat_large | 0 | 60 | 21.483 | 3.877 | 0.501 |
| | 1 | 30 | 23.200 | 1.730 | 0.316 |
| allocat_small | 0 | 60 | 8.517 | 3.877 | 0.501 |
| | 1 | 30 | 6.800 | 1.730 | 0.316 |

* Constraint_dummy = 0 (1) for auctions with lower (highest) capacity constraint.

The *t*-tests for the equality of the means reported in Table 9 show that all the differentials were significant.

**Table 9.** Levene's test for the equality of variances and the *t*-test for the equality of the means of the outcomes of the auction with the highest and lower capacity constraints.

| | Levene's Test Equality of Variances | | | *t*-Test Equality of Means | | | |
|---|---|---|---|---|---|---|---|
| Variable | Assumption | F | Sig. | t | df | Sig. 2 Tail | Mean Diff. |
| UP | Equal variances assumed | 3.797 | 0.055 | −4.310 | 88 | 0.000 | −9.960% |
| | Equal variances not assumed | | | −4.686 | 73 | 0.000 | −9.960% |
| payoff | Equal variances assumed | 10.709 | 0.002 | −3.885 | 88 | 0.000 | −51.719 |
| | Equal variances not assumed | | | −4.819 | 88 | 0.000 | −51.719 |
| freq | Equal variances assumed | 0.658 | 0.420 | 1.897 | 88 | 0.061 | 4.970% |
| | Equal variances not assumed | | | 1.977 | 65 | 0.052 | 4.970% |
| error | Equal variances assumed | 10.495 | 0.002 | 2.149 | 88 | 0.034 | 0.667 |
| | Equal variances not assumed | | | 2.577 | 87 | 0.012 | 0.667 |
| allocat_large | Equal variances assumed | 12.944 | 0.001 | −2.308 | 88 | 0.023 | −1.717 |
| | Equal variances not assumed | | | −2.900 | 87 | 0.005 | −1.717 |
| allocat_small | Equal variances assumed | 12.944 | 0.001 | 2.308 | 88 | 0.023 | 1.717 |
| | Equal variances not assumed | | | 2.900 | 87 | 0.005 | 1.717 |

The results confirm that imposing the highest level of capacity constraint on investors resulted in a significantly greater contribution of private information to price discovery. The most restrictive auctions exhibited a significantly lower pricing error, a significantly lower incidence of free riding, and a significantly higher share allocation to the most informed large investors. Moreover, the process of price discovery is more reliable in the most restrictive auctions as indicated by the significantly lower standard deviation of both the pricing error and the share allocations to large investors. The standard deviation of free riding frequency was also lower in the most restrictive auctions but the difference was not significant.

The less restrictive auctions offered investors an average payoff that was close to zero (1.920) which was significantly less than the average payoff of 53.639 of the most restrictive auctions. The fact that the standard deviation of the payoff of the less restrictive auctions was also significantly higher (68.969) than that of the most restrictive auctions (32.825) makes participation in the less restrictive auctions much riskier for investors given the relatively high probability of generating a negative return[10]. This lowers the probability that informed investors will reliably contribute their private information in price discovery.

The implication of the findings on the effect of imposing capacity constraints is similar to the findings on the effect of including discounts. IPO firms face a trade-off between underpricing and the reliability of IPO price discovery. By choosing the most restrictive auctions, the reliability significantly improves but is accompanied by a significantly higher level (10.245% versus 0.285%) of underpricing but also significantly more predictable (standard deviation of 8.607% versus 11.085%) underpricing.

Table 10 shows the average underpricing, payoff, frequency of free riding, pricing error allocations to large and small investors of auctions that offered discounts and in which the highest capacity constraint was imposed upon investors (the 20 auctions of treatments 6 and 9) and of the other auctions with a discount that were less restrictive (the 40 auctions of treatments 4, 5, 7, and 8). Imposing the highest capacity constraint lowers the average pricing error as well as the frequency of free riding and increases the allocations to large more informed investors but also increases underpricing. Moreover, all the auction outcomes with the highest capacity constraint exhibited a lower standard deviation, making the auction outcomes more predictable. However, we cannot reliably test if these differences between the two groups' auction outcomes were significant due to the small sample size (20) of the auctions with both a discount and a highest capacity constraint.

**Table 10.** Outcomes of the auctions with a discount in combination with the highest capacity constraint or with lower capacity constraints.

| Constraint_Dummy * | | N | Mean | Std. Deviation | Std. Error Mean |
|---|---|---|---|---|---|
| UP | 0 | 40 | 4.137% | 9.519% | 1.505% |
| | 1 | 20 | 12.958% | 8.974% | 2.007% |
| payoff | 0 | 40 | 27.630 | 55.013 | 8.698 |
| | 1 | 20 | 65.458 | 28.714 | 6.421 |
| freq | 0 | 40 | 52.778% | 11.667% | 1.845% |
| | 1 | 20 | 48.333% | 10.370% | 2.319% |
| error | 0 | 40 | 1.525 | 1.377 | 0.218 |
| | 1 | 20 | 1.000 | 0.858 | 0.192 |
| allocat_large | 0 | 40 | 22.550 | 3.121 | 0.493 |
| | 1 | 20 | 23.500 | 1.732 | 0.387 |
| allocat_small | 0 | 40 | 7.450 | 3.121 | 0.493 |
| | 1 | 20 | 6.500 | 1.732 | 0.387 |

* Constraint_dummy = 0 (1) for auctions with lower (highest) capacity constraint.

---

[10] We find no significant difference between the outcomes of the auctions with a low versus mid capacity constraints. More importantly, given the payoff of respectively −1.44 (SD 64.374) and 5.28 (SD 74.253), both these auctions did not provide a reliable incentive for investor participation.

## 4. Discussion

Our results confirm Hypothesis 1, that offering informed investors a discount on the auction clearing price—while significantly improving the accuracy and reliability of IPO price discovery—leads to underpricing. We found an average underpricing of 7.078% (SD 10.169%) for auctions with a discount of 10%. There are two markets where discounts on uniform auction clearing prices were offered to investors and where empirical studies have been performed on the resulting underpricing and accuracy and reliability of price discovery: France and the US[11].

Derrien and Womack (2003) found that price discovery has been more accurate and reliable in French IPO auctions than in book building while also leading to less underpricing and underpricing variability. They report an average underpricing of 9.68% (SD 12.25%) for auctions and significantly higher underpricing of 16.89% (SD 24.49%) for a sample of comparable book built IPOs. In the US, Lowry et al. (2010) reported an average underpricing of 1.5% (SD 10.1%) for auctions which is significantly less than the average underpricing of 22.0% (SD 47.6%) of a matched sample of book-built IPOs. We note that the standard deviation of underpricing that we have found is similar to that of IPO auctions executed in practice. Since the average discount applied in the auctions was not reported by Lowry et al. (2010) or by Derrien and Womack (2003), we cannot compare the underpricing levels found in our experiment with that of the IPOs that were auctioned in practice.

Nevertheless, the conclusion that offering discounts in auctions improves IPO price discovery is supported by the evidence. We note that—since the clearing price is not reported in book building—we cannot directly test the effect of offering discounts on the accuracy and reliability of price discovery in that IPO mechanism. Consequently, we can only compare underpricing levels and the underpricing variability of book-built versus auctioned IPOs. If researchers would be allowed access to the bidding books of investment banks that have used book building for the pricing and allocation of IPOs, this would enable a more detailed analysis of the impact of discounts on the efficiency of price discovery.

Our results confirmed Hypothesis 2 that imposing capacity constraints leads to more accurate and reliable price discovery but increases underpricing. We found that it significantly lowered the pricing error from 1.667 (SD 1.580) to 1.000 (SD 0.871) but increased underpricing from 0.285% (SD 11.085%) to 10.245% (SD 8.607%). Schnitzlein and Shao (2013) reported that imposing the highest capacity constraint while holding the number of available shares constant, significantly lowered both pricing error from 1.89 (SD 1.78) to 1.10 (SD 1.44) and auction revenue[12] by 29.59. We also found that imposing the highest capacity constraint significantly lowered the frequency of free riding which—similar to the findings of Schnitzlein and Shao (2013)—nevertheless remains stubbornly high. We recommend future experiments that may reveal other ways to combat this strategic bidding behavior.

The results obtained indicate that the most capacity restricted auctions that also offer investors a discount on the market clearing price are likely to produce the most reliable price discovery and consequently, the most predictable auction outcome. The findings confirm Hypothesis 3 of this study that offering a discount and imposing capacity constraints significantly improves IPO valuation accuracy and reliability but at the expense of more underpricing.

There are indications that a discount of 8% may suffice to incentivize investors to reliably contribute to the price discovery process. The resulting underpricing (and its variability) is likely to be significantly lower than if book building would be used to price and allocate IPOs.

An important implication for economic policy makers who are searching for ways to improve the functioning of the IPO market is that they are more likely to achieve their objectives by making

---

[11]   According to Jagannathan et al. (2015) who surveyed internationally used auction mechanisms, both the US and France used uniform price auctions with discounts on the auction clearing price.

[12]   Schnitzlein and Shao (2013) did not report underpricing but instead auction revenue—the number of shares sold times the auction clearing price.

structural changes in the way IPOs are priced and allocated than by introducing legislation that does not alter the IPO mechanism itself.

Another implication of this study is that it provides academic researchers with guidance for designing an optimal IPO mechanism. We note, however, that the results should be interpreted with caution. Firstly, the results were generated in a laboratory setting. Secondly, the experiments measured the impact of only two potential variables—at just three discrete levels—on IPO price discovery.

Designing the optimal IPO mechanism most likely requires a much more fine-grained and complex analysis that takes into account a number of factors that were not included in this study. Firstly, there is the strong empirical support in the IPO literature that the degree of information asymmetry between investors that are able and willing to participate in price discovery is a function of what is commonly referred to as ex ante uncertainty in the literature (Ljungqvist 2007). While studies have identified specific IPO firm characteristics that are suitable as proxies for ex ante uncertainty, there is no comprehensive model that relates IPO firm characteristics to ex ante uncertainty. Such a model would be useful todetermine the extent to which investors need to be incentivized and restricted in IPO price discovery. Given the wide variety and evolving nature of IPO company characteristics—an important one of which is the industry the firm is from (Benveniste et al. 2003)—it may require the application of data analytics on an up-to-date database to determine the degree of ex ante uncertainty surrounding the valuation of an IPO candidate.

Secondly, it is very likely that there are considerable differences in the degree to which institutional investors are able to contribute their private information to IPO price discovery. IPO firms are often young companies that depend on relatively unproven technology and business models. The accurate valuation of such firms requires specialized knowledge and experience that is most likely not evenly distributed among institutional investors. Determining the relative potential contribution to a specific IPO firms' price discovery for each of the thousands of institutional investors could again be objectivized and quantified by using data analytics. This in turn would allow the detailed design of the optimal IPO mechanism.

Finally, the experimental results of imposing constraints indicate that these have a significant impact on the accuracy and reliability of IPO price discovery. However, the incidence of free riding remains stubbornly high. Future experiments—probably involving machine learning to reliably predict the bidding behavior based on the historical patterns of bidding—should focus on the auction designs that combat this opportunistic behavior that negatively impacts the efficiency of price discovery.

Technological innovation in the IPO market—through the application of recent advances in data science, experimental economics and artificial intelligence—allows for the optimization of IPO mechanisms and crowdfunding platforms which, in turn, improves access to equity required for entrepreneurial finance.

**Funding:** This research received no external funding.

**Conflicts of Interest:** The author declares no conflict of interest.

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
