# Peer review of "Towards an Optimal IPO Mechanism"

_jrfm, doi:10.3390/jrfm13060115_

Round 1

Reviewer 1 Report

typo line 57

et al. vs et al (or a typo?) see lines 57 and 86

Author Response

I followed the suggestion of reviewer 1 and changed "et al" into "et al." in both footnote 12 and line 87 of the  manuscript. However, I could not find the typo in lines 57 and 86.

Reviewer 2 Report

  • In the abstract of the paper, it is necessary to highlight the purpose of the study. Also, in the abstract, describe the main methods applied (just briefly).
  • The introduction of the paper lacks a clearer explanation why is the topic of the paper important for the research.
  • The introduction should contain the main hypothesis of research and/or the key research questions.
  • In the last paragraph of the introduction you mentioned: “Starting from the standard auction, we first measure the effect of incentivizing informed investors with discounts on valuation accuracy, reliability and underpricing. We also measure the effect of imposing capacity constraints on valuation accuracy, reliablity and underpricing. Third, we measure the combined effect of offering a discount and imposing capacity constraints.”     These are research activities that will be used to achieve the aim of research. But, what is the main aim of your research? You put in the first sentence of the last paragraph: “The objective of this paper is to shed light on the nature of the trade-off by conducting a controlled experiment.”  It is too generally! -What does it mean “to shed light on”!?    Please, try to be more concrete and clearly emphasize the main aim of your paper!
  • In the end of the introduction you explain your scientific contribution: “Our work is most closely related to the experiment of Schnitzlein et al. (2019) who measure the effect of excluding retail investors on valuation accuracy, reliablity and underpricing in standard auctions. Our contribution is that we extend their work by measuring the effects on auction outcomes by varying incentives and capacity constraints for institutional investors.”    It is very poor explanation of the contribution of your research to the science.    The author/s need to do a better job of situating this work in the existing literature and clearly stating the paper’s novel contribution to that literature.
  • The scientific contribution of the paper should stem from the main aim of the paper. Generally speaking, the paper lacks better connection and compatibility between the main aim of the paper, hypothesis, research methodology and contribution to the science.
  • In the chapter Discussion, among others, you discussed the results obtained and how they can be interpreted in perspective of previous studies. And that is fine. However, in this chapter, try to discuss the findings and their implications in the broadest context possible, including working hypothesis/research questions etc., in order to highlight new scientific contributions of your paper.
  • In order to improve research design it would be recommendable to add the chapter 5. Conclusion. For that purpose, you can use the content of the second part of the chapter Discussion, beginning with the sentence: All in all, our results...     You can begin the conclusion with: The results obtained indicate that the most capacity restricted auctions that also offer investors a discount on the market clearing price are likely to produce the most reliable price discovery and, consequently, most predictable auction outcome...
  • In the conclusion of the paper it is necessary to connect obtained results with the main aim of the research and the main hypothesis/research questions.

Author Response

Thank you for your review.

  • In the abstract of the paper, the purpose of the study is highlighted in lines 13 and 14: “The purpose of this study is to measure the impact of two key variables on this trade-off: capacity restraint and discount on the auction clearing price.” Also, in lines 14 and 14 of the abstract, we describe the main methodology applied (just briefly) as: “Using controlled experiment methodology in multi-unit uniform price auctions we find that…”
  • A clearer explanation why is the topic of the paper important for the research is added in lines 73-77. “The mixed evidence suggests that there is room for improvement on a more fundamental level: changing the IPO process itself. This study contributes to the research dedicated to the optimisation of the IPO mechanism in order to improve access to funding for young innovative companies that are important contributors to economic growth and employment. ”
  • In the introduction we added the hypotheses, including the main hypothesis in lines 118-127:“The purpose of this study is to measure the impact of two key variables on this trade-off: capacity restraint and discount on the auction clearing price by conducting a controlled experiment. We hypothesize that discounts significantly improve IPO valuation accuracy and reliability but at the expense of more underpricing (hypothesis 1). Starting from the standard auction, we first measure the effect of incentivizing informed investors with discounts on valuation accuracy, reliability and underpricing. Second, we hypothesize that imposing capacity constraints significantly improve IPO valuation accuracy and reliability but at the expense of more underpricing (hypothesis 2). Therefore, we also measure the effect of imposing capacity constraints on valuation accuracy, reliablity and underpricing. Third, we measure the combined effect of offering a discount and imposing capacity constraints to test if there is a combined effect of the two key variables on valuation accuracy, reliability and underpricing (hypothesis 3; main hypothesis).”
  • The main aim of the research is given in lines 118-119: “The purpose of this study is to measure the impact of two key variables on this trade-off: capacity restraint and discount on the auction clearing price by conducting a controlled experiment.”
  • The paper’s novel contribution to the literature is stated in lines 132-136: ”The combination of imposing capacity constraints and offering discounts in auctions has a positive effect on valuation accuracy and reliability but at the cost of additional underpricing. This study makes an important and novel contribution to the academic literature about the optimalisation of IPO mechanisms by measuring the impact of two key variables on IPO valuation accuracy, reliability and underpricing.”
  • Improved the connection between the aim of the paper, the hypotheses and the research methodology in lines 118-127: “The purpose of this study is to measure the impact of two key variables on this trade-off: capacity restraint and discount on the auction clearing price by conducting a controlled experiment. We hypothesize that discounts significantly improve IPO valuation accuracy and reliability but at the expense of more underpricing. Starting from the standard auction, we first measure the effect of incentivizing informed investors with discounts on valuation accuracy, reliability and underpricing. Second, we hypothesize that imposing capacity constraints significantly improve IPO valuation accuracy and reliability but at the expense of more underpricing. Therefore, we also measure the effect of imposing capacity constraints on valuation accuracy, reliablity and underpricing. Third, we measure the combined effect of offering a discount and imposing capacity constraints to test if there is a combined effect of the two key variables on valuation accuracy, reliability and underpricing.” and contribution to the science in lines 132-136: ”The combination of imposing capacity constraints and offering discounts in auctions has a positive effect on valuation accuracy and reliability but at the cost of additional underpricing. This study thus makes an important and novel contribution to the academic literature about optimal IPO mechanisms by measuring the impact of both key variables on IPO valuation accuracy, reliability and underpricing by using a controlled experiment.”
  • In the chapter Discussion we discuss the findings, in the context of:
    • hypothesis 1 in lines 335-337: “Our results confirm hypothesis 1 that offering informed investors a discount on the auction clearing price – while significantly improving the accuracy and reliability of IPO price discovery – leads to underpricing.”
    • hypothesis 2 in lines 362-363: ” Our results confirm hypothesis 2 that imposing capacity constraints leads to more accurate and reliable price discovery but increases underpricing.”
    • hypothesis 3 in lines 372-374: “In line with hypothesis 3, our results indicate that the most capacity restricted auctions that also offer investors a discount on the market clearing price are likely to produce the most reliable price discovery and, consequently, most predictable auction outcome.”

In the chapter Discussion we discuss the implications of the findings in a broad context in lines 377-382: “An important implication for economic policy makers who are searching for ways to improve the functioning of the IPO market is that they are more likely to achieve their objectives by making structural changes in the way IPOs are priced and allocated than by introducing legislation that does not alter the IPO mechanism itself. Another implication of this study is that it provides academic researchers guidance for designing an optimal IPO mechanism.”

  • We added the chapter 5 Conclusion beginning with the sentence in lines 416-418: “The results obtained indicate that the most capacity restricted auctions that also offer investors a discount on the market clearing price are likely to produce the most reliable price discovery and, consequently, most predictable auction outcome.”
  • In the conclusion we connect the obtained results with the main aim of the research and the main hypothesis in lines 418-420: “The findings confirm the main hypothesis of this study that offering a discount and imposing capacity constraints significantly improve IPO valuation accuracy and reliability but at the expense of more underpricing.”

Reviewer 3 Report

I state that

This article investigate the issue of optimization of IPO mechanisms in the IPO market using an application of electronic price auctions for the purpose to improve access to equity for IPO firms.

I recommend to consider and make clearly: the title of the article, in abstract - clearly formulate the main aim of this article and used research methodology/methods, in Introduction, too, fill in key words, too- to reflect the area of the research, (IPO and initial public offering – still the same), explain value added of this article with focus on effects and risks, in Conclusion.

Author Response

Thank you for your review!

In line 2 the title of the article has been changed to better reflect the contents of the paper: “Towards an optimal IPO mechanism”

In the abstract the purpose of this article is added and used research methodology in lines 13-15: “The purpose of this study is to measure the impact of two key variables on this trade-off: capacity restraint and discount on the auction clearing price. Using controlled experiment methodology in multi-unit uniform price auctions…”

the purpose of this article is added and used research methodology are also added in the Introduction in lines 118-119: “The purpose of this study is to measure the impact of two key variables on this trade-off: capacity restraint and discount on the auction clearing price by conducting a controlled experiment.”

In line 24 I have deleted “initial public offering” for list of Keywords as IPO and initial public offering are the same.

In lines 132-136 the value added of this article is explained: ”The combination of imposing capacity constraints and offering discounts in auctions has a positive effect on valuation accuracy and reliability but at the cost of additional underpricing. This study makes an important and novel contribution to the academic literature about the optimalisation of IPO mechanisms by measuring the impact of two key variables on IPO valuation accuracy, reliability and underpricing.”

In addition, we discuss the implications of the findings in a broad context in lines 377-382: “An important implication for economic policy makers who are searching for ways to improve the functioning of the IPO market is that they are more likely to achieve their objectives by making structural changes in the way IPOs are priced and allocated than by introducing legislation that does not alter the IPO mechanism itself. Another implication of this study is that it provides academic researchers guidance for designing an optimal IPO mechanism.”

Round 2

Reviewer 2 Report

  • You did not accept and implement my comment related to the main aim of your research. Obviously, we did not understand each other when we talk about the difference between the main aim of the research and activities to achieve the main aim.     Hint. You can write: “The main aim of this paper is to gain new scientific knowledge about …”
  • Expand the Conclusion section (chapter 5) with the main findings of your research (several sentences without stating the numbers). Or move the two sentences you wrote in the conclusion to the Discussion chapter.

Author Response

Thank you for your suggestions. 

I have added in the Abstract, lines 13-15: "The main aim of this paper is to gain new scientific knowledge about this trade-off by measuring the impact of two key variables on this trade-off: capacity restraint and discount on the auction clearing price. "

I have added in lthe Introduction, lines 118-120: "The main aim of this paper is to gain new scientific knowledge about this trade-off by using a controlled experiment to measure the impact of two key variables on this trade-off: capacity restraint and discount on the auction clearing price. "

I followed the suggestion to  move the two sentences that I wrote in the conclusion to the Discussion chapter, lines 383-388: "The results obtained indicate that the most capacity restricted auctions that also offer investors a discount on the market clearing price are likely to produce the most reliable price discovery and, consequently, most predictable auction outcome. The findings confirm the main hypothesis of this study that offering a discount and imposing capacity constraints significantly improve IPO valuation accuracy and reliability but at the expense of more underpricing."

Round 3

Reviewer 2 Report

The revised version of your manuscript has been significantly improved. The paper is written in a scientifically justifiable and consistent way. The approach and research methodology are determined according to the aim of the paper. The obtained results are scientifically justifiable, consistent and comparable, and they offer scientific contribution.